# A Taxonomic Survey of Female Oviducal Glands in Chondrichthyes: A Comparative Overview of Microanatomy in the Two Reproductive Modes

**DOI:** 10.3390/ani11092653

**Published:** 2021-09-09

**Authors:** Martina Francesca Marongiu, Cristina Porcu, Noemi Pascale, Andrea Bellodi, Alessandro Cau, Antonello Mulas, Paola Pesci, Riccardo Porceddu, Maria Cristina Follesa

**Affiliations:** 1Dipartimento di Scienze della Vita e dell’Ambiente, Università degli Studi di Cagliari, Via Tommaso Fiorelli 1, 09126 Cagliari, Italy; pascalenoemi3@gmail.com (N.P.); abellodi@unica.it (A.B.); alessandrocau@unica.it (A.C.); amulas@unica.it (A.M.); ppesci@unica.it (P.P.); riccardo.porceddu@unica.it (R.P.); follesac@unica.it (M.C.F.); 2Consorzio Nazionale Interuniversitario per le Scienze Mare (CoNISMa), Piazzale Flaminio 9, 00196 Roma, Italy

**Keywords:** oviducal gland, histology, sperm storage, oviparity, viviparity

## Abstract

**Simple Summary:**

The oviducal gland (OG) is a specialized region of the reproductive female system in cartilaginous fish located in the anterior oviduct. Its biological importance is closely related to the reproductive modalities of these species, and its basic function is the production of the egg jellies, the tertiary envelope formation (egg case in oviparous and candle case in viviparous) and sperm storage. Since knowledge on the overall process of Chondrichthyes reproduction is still scarce, in this study we conducted morphological and morphometrical analysis on the OGs belonging to several cartilaginous fish displaying two different reproductive modalities (oviparity and viviparity). Moreover, we paid particular attention to the fate of spermatozoa in the female reproductive tract, which would be useful to better understand the ecology and population dynamics of these species.

**Abstract:**

Oviducal glands (OGs) are distinct expanded regions of the anterior portion of the oviduct, commonly found in chondrichthyans, which play a key role in the production of the egg in-vestments and in the female sperm storage (FSS). The FSS phenomenon has implications for understanding the reproductive ecology and management of exploited populations, but little information is available on its taxonomic extent. For the first time, mature OGs from three lecithotrophic oviparous and four yolk-sac viviparous species, all considered at risk from the fishing impacts in the central western Mediterranean Sea, were examined using light microscopy. The OG microanatomy, whose morphology is generally conserved in all species, shows differences within the two reproductive modalities. Oviparous species show a more developed baffle zone in respect to viviparous ones because of the production of different egg envelopes produced. Among oviparous species, *Raja polystigma* and *Chimaera monstrosa* show presence of sperm, but not sperm storage as observed, instead, in *Galeus melastomus* and in all the viviparous sharks, which preserve sperm inside of specialized structures in the terminal zone.

## 1. Introduction

Chondrichthyes, as well as being the only anamniote vertebrate class that exclusively employs internal fertilization during reproduction [1], have several features in common (low growth rate, delayed maturity and long gestation), all resulting in low reproductive potential that makes them highly vulnerable to overfishing (e.g., [2,3]). Chondrichthyan fish exhibit two main modes of reproduction, spanning both oviparity and viviparity. Oviparity (i.e., eggs are enclosed within an egg case and deposited in the sea) is restricted to the orders Carcharhiniformes, Heterodontiformes, Orectolobiformes, Rajiformes and Chimaeriformes, while viviparity is prevailing in all other chondrichthyan orders [4]. Viviparous species are further subcategorized as lecithotrophic and matrotrophic [5,6]. Depending on the pattern of embryonic nutrition, in the lecithotrophic mode, embryos do not receive any maternal nourishment, which is granted, instead, by a yolk-sac (yolk reserves); in the matrotrophic mode, embryos receive nourishment through ingestion of lipids or mucous (histotrophy) produced by the uterine walls of the mother [5,6]. Oophagy and intrauterine cannibalism (those that eat eggs or other embryos) are also enclosed in the matrotrophic reproductive mode [5,6].

The morphology of the female reproductive system is generally conserved, and the genital duct displays a variety of specializations linked to the different reproduction modes [7]. All Chondrichthyes show a glandular enlargement of the oviduct known as oviducal gland (OG) interposed between the oviduct and the uterus, except for the Narcinidae family in which the OG is lacking [8] and some species of Myliobatiformes which have a vestigial OG [9,10]. The OG is responsible mainly for producing secondary and tertiary egg coats [11], revealing four distinct secretory zones (e.g., [11,12,13,14,15,16]): club and papillary zones that produce mucus surrounding the fertilized ovum in the early stages of embryogenesis, the baffle zone that forms the tertiary envelope of the egg (egg case in oviparous and candles in viviparous species) and the terminal zone where the eggs’ ornamentations are produced and the sperm can be stored (e.g., [16,17,18,19,20,21]). A small number of chondrichthyan species, such as the yellow stingray *Urobatis jamaicensis* (Cuvier 1816), do not produce an egg envelope, and the baffle zone is modified accordingly [17]. The size and structural complexity of the OG, as well as the reproductive relevance of each glandular region, are correlated with the mode of reproduction (e.g., [5,22,23,24]). The shape of the OG is also highly variable at the gross level, but as a general rule, oviparous species have the largest OGs among chondrichthyans, since they produce not only the jellies that envelop the egg, but also an external capsule and all its ornamentations (e.g., [15,25]). Conversely, viviparous sharks show a uniform breakdown of OG volume due to a small dimension of the baffle zone (in Triakidae members the OG is virtually indistinguishable) (e.g., [7]). 

Female sperm storage is the extended maintenance of viable sperm prior to its use in fertilization and likely occurs in most internally fertilizing animals [26]. Female sperm storage has been described in nematodes, annelids, squids and arthropods and in all major groups of vertebrates, including amphibians, cartilaginous fishes (sharks and rays), bony fish, squamates, reptiles, birds and mammals [26]. Female sperm storage makes delayed fertilization possible and facilitates a temporal separation between copulation and fertilization, even if not all stored sperm is used for this purpose. It can be viewed as an evolutionarily conserved strategy to ensure reproductive efficiency and increase the probability of insemination of nomadic species or in those with low population density [27], impacting many aspects of their biology, including life histories, mating systems, cryptic female choice and sperm competition, and ultimately may lead to sexual conflict [26]. Sperm storage has been found in the terminal zone of multiple chondrichthyans, mainly sharks and holocephalans (e.g., [7,16,18,19,20,28]). 

Of the 88 chondrichthyan species recorded in the Mediterranean Sea [29], more than half (at least 53%; [30]) are threatened because of overfishing and are classified by the International Union for Conservation of Nature classification as vulnerable, endangered or critically endangered [30]. In this sense, understanding the overall process of reproduction would be useful for assessing the population status of these species [31] by investigating male–female interactions, physiology, biochemistry and anatomy [7,18], thus providing scientifically reliable advice for understanding their ecology and population dynamics [7,18]. Although chondrichthyans tend to be apex predators and fulfill many key roles in their ecosystems [32], there has been little global research on basic biology and population composition for many species [33]. 

For these reasons, the goal of this study is to provide a broader taxonomic survey of female oviducal glands across different reproductive modes present in some demersal chondrichthyans inhabiting the Mediterranean Sea. More specifically, we investigated (i) the microanatomy and the carbohydrate distribution pattern in the different zones of the OGs through morphological, histological and histochemical observations for three oviparous species and four yolk-sac viviparous ones and (ii) sperm storage and sperm distribution in the two reproductive modes. 

## 2. Materials and Methods

### 2.1. Sampling

Female specimens of seven chondrichthyan species were collected between 2012 and 2020 around Sardinian waters (central western Mediterranean) during the Mediterranean International Trawl Survey (MEDITS; [34]) and commercial hauls along with data collected monthly from commercial landings through the Data Collection Framework (European Union Regulation 199/2008). Collection and handling of animals took into account the ethical and welfare considerations approved by the ethics committee of the University of Cagliari (Sardinia, Italy). 

For each individual, the total length (TL, in centimeters) and the oviducal gland width (OGW, in millimeters considering the maximum breadth) were recorded. Only for the holocephalan *Chimaera monstrosa*, the anal length (AL, in centimeters) was taken. Maturity stages were established following the scale for oviparous and viviparous elasmobranchs [35]. According to these scales, oviparous females were classified into six stages on the basis of ovary structure and OG and uterus dimension and texture as follows: stage 1, immature (F1); stage 2, developing (F2); stage 3a, spawning capable (F3A); stage 3b, actively spawning (F3B); stage 4a, regressing (F4A); and stage 4b, regenerating (F4B). Instead, viviparous females were classified into seven stages: stage 1, immature (F1); stage 2, developing (F2); stage 3a, capable of reproducing (F3A); stage 3b, early pregnancy (F3B); stage 3c, mid-pregnancy (F3C); stage 3d, late pregnancy (F3D); stage 4a, regressing (F4A); and stage 4b, regenerating (F4B). 

### 2.2. Histological Procedures

For the histological analysis and, in particular for the analysis regarding the sperm storage, were used only mature OGs fully developed from stage 3a to 4b. A subsample of five OGs for each maturity stage (when available) was processed for histological analysis. Whole OGs were first fixed in 5% buffered formaldehyde (0.1 mol L^−1^, pH 7.4) for a maximum period of 48 h. They were then dissected by cutting through the center from the oviduct to the uterus, which provided a piece of tissue representing the sagittal plane of the gland. The tissues were embedded in a synthetic resin (GMA, Technovit 7100, Bio-Optica, Milan, Italy) following routine protocols and sectioned at 3.5 μm with a rotating microtome (ARM3750, Histo-Line Laboratories, Pantigliate, Italy). Slides were stained with hematoxylin and eosin (H&E) for standard histology and with periodic acid–Schiff (PAS) and Alcian blue (AB) in combination to assess the production of neutral and sulfated acid mucins [36]. Subsequently, sections were dehydrated in graded ethanol (96–100%), cleared in Histolemon (Carlo Erba Reagents, Cornaredo, Italy) and mounted in resin (Eukitt, Bio-Optica). Selected sections were observed and photographed using a Nexcope NE600 optical microscope equipped with a digital camera (MD6iS) at different magnifications (40×, 100× and 400×) and edited with Adobe Photoshop CS6 /(http://www.adobe.com/products/photoshop.html (22 July 2021)).

### 2.3. Macroscopic and Microscopic Measurements and Statistical Analysis

Analysis of variance (ANOVA, multiple range test) was used to test for statistical differences in OG width among all the maturity stages [37] using the software Stat-graphics Centurion XVI to test differences between the several reproductive cycle phases. 

To evaluate the surface (expressed in mm^2^ and in %) occupied by each OG zone (club, papillary, baffle and terminal), ImageJ software [38] was used. Each zone was measured considering the entire area covered by tubules and lamellae, excluding the connective tissue that borders the OG (Appendix A). 

TpsDig software [39] was used to collect the detailed measurements of the OG and check differences between the four zones. The dimension of gland tubules was evaluated measuring their maximum length (Appendix A). Lamellae from each zone were counted and measured starting from the base towards the lumen (Appendix A).

## 3. Results

In Table 1, the analyzed species, grouped based upon their mode of reproduction (oviparous and yolk-sac viviparous), are presented.

### 3.1. Macroscopic Development of the OG

Comparing the morphology of the oviducal glands of the oviparous species, they showed different shapes. *C. monstrosa* and *G. melastomus* glands had an oval shape (Appendix A), while *R. polystigma* glands had a heart shape (Appendix A).

Considering the maturation cycle of each species, the OGs become larger when females attained maturity, reaching the maximum values at stages 3A and 3B with a slight decline in the postspawning phase in all examined species (Figure 1). Statistically significant differences during the evolution of the OGs of all species were found (*C. monstrosa*, ANOVA, F-ratio = 167.40, statistically significant differences between all stages except stages 3A–3B, *p*-value = 0; *G. melastomus*, ANOVA, F-ratio = 97.40, statistically significant differences between all stages except stages 3A–4A, 3A–4B, 4A–4B, *p*-value = 0; *R. polystigma*, ANOVA, F-ratio = 379.48, statistically significant differences between all stages except stages 3B–4A, *p*-value = 0). 

The oviducal glands of all viviparous species had a barrel shape with considerable lateral extension visible (Appendix A). In *O. centrina*, a dark stripe in the external part of the gland was visible (Appendix A). The OG of *C. uyato*, *D. licha* and *O. centrina* reached the maximum development in capable of reproducing females (stage 3A) with a subsequent decline in the maternal stages (Figure 1, ANOVA, *C. uyato*, F-ratio = 1340.41, statistically significant differences between all stages, *p*-value = 0; *D. licha*, F-ratio = 60.26, statistically significant differences between all stages, *p*-value = 0; *O. centrina*, F-ratio = 13.68, statistically significant differences between all stages except stages 3B–3C, 3C–3D, 4A–4B, *p*-value = 0.0001). Only few specimens of *H. perlo* were caught during the sampling period, and for this reason, it was not possible to perform a statistical analysis. 

### 3.2. Microscopic Development of the OG

#### 3.2.1. Oviparous Species

##### *Chimaera* *monstrosa*

On the basis of AB-PAS staining, each different zone displayed unique staining affinities (Figure 2A).

The club zone (Table 2) showed the club-shaped lamellae (Table 3), characterized by a simple, ciliated and columnar epithelium (Figure 3A). Tubules were very elongated near the lumen and were composed of ciliated columnar cells with basal nuclei and secretory cells containing a mixture of AB+ and PAS+ secretory material (Figure 3B, Table 4). The papillary zone, which occupied a similar surface to the club zone (Table 2), showed sever-al digitiform lamellae (Table 3; Figure 3C). As with the club zone, the tubular glands were elongated near the lamellae (Figure 3C), while in the deep recesses, they were more rounded and showed less affinity for dyes (Figure 3D). Secretory cells of the caudal-most papillary tubules (Figure 3E), adjacent to the baffle zone, produced a less AB+/PAS+ secretory material than the papillary zone. The baffle zone, the most conspicuous part of the OG (Table 2), consisted of 34 apical expanded lamellae (Table 3) that alternated with transversal grooves (Figure 3F). The glandular duct ended in a pair of spinnerets, then passing to the transverse grooves (Figure 3F). Tubular glands, PAS/AB-negative (Figure 3G), had cells with round cytoplasmatic granules, basal nucleus and evident nucleolus. The terminal zone was the smallest part of the gland (Table 2), and it had no lamellae but had crypts where the gland ended (Figure 3H). In this zone, the tubular glands (Table 3) mainly had PAS-positive mucous cells (Figure 3I). Accumulation of secretions in the lumen of secretory glands and between lamellae was observed in all maturity stages (from mature to regenerating females), being more evident in the baffle and terminal zones.

##### *Galeus* *melastomus*

The blackmouth catshark OG displayed a similar structure compared to *C. monstrosa* (Figure 2B). The club zone presented very developed lamellae in width rather than in length (Figure 4A, Table 3). The granules of the secretory cells were PAS+ and AB+ (neutral and sulfated acid mucins) (Figure 4B, Table 4). The papillary zone, smaller than the club zone (Table 2) and characterized by few lamellae (Table 3, Figure 4C), showed an affinity only for PAS (Figure 4D). Tubules, surrounded by a basement membrane, are simple, and secretory materials produced inside are PAS+ (Figure 4D), identifying the production of neutral mucopolysaccharides. Differently from *C. monstrosa*, *G. melastomus* did not show the caudal-most papillary zone. The baffle zone is very large (Table 2) causing the gland to be much larger, with an increased luminal profile (Figure 4E, Table 3). The secretory ducts open into the lumen through the spinneret region, composed of two baffle plates. The baffle plates are surrounded by another pair of large folds, the plateau projections, lining the transverse groove (Figure 4E). Secretory tubules did not react to any of the special staining techniques (PAS-, AB-) (Figure 4F, Table 4), while the cells near the lamellae had a weak affinity for PAS (PAS+). The structural organization of the terminal zone (Table 2) displayed no lamellae and a regular surface epithelium with elongated tubular glands opening into the lumen (Figure 4G). The elongated secretory tubules (Table 4) consisted mainly of mucous secretions with large vesicle cells intensely blue (AB+) (Figure 4H).

##### *Raja* *polystigma*

In females spawning capable of and actively spawning, the OG was fully developed with the four zones clearly recognizable (Figure 2C), enclosed by a serosa of dense connective tissue. The luminal surface of the lamellae of all zones was composed of a simple ciliated columnar epithelium supported by loose connective tissue (lamina propria). The club zone (Table 2) had the characteristic club-shaped lamellae (Figure 5A, Table 3), and the glandular epithelium tubules (Figure 5B, Table 4) were composed of ciliated and secretory cells. CZ reacted strongly to PAS/AB in the area near the lamellae (deep purple color) and slightly less in the deep recesses (purple color). The papillary zone, recognizable by the digit-shaped lamellae (Figure 5C, Table 3), had simple or occasionally ramified tubules and secretory tubules producing mucins PAS+/AB+ (Figure 5D, Table 4). Adjacent to the baffle zone, a row of secretory tubules (caudal-most papillary zone) (Figure 5E) reacted slightly less than the papillary zone. The BZ, the most conspicuous and extensive segment (Table 2), did not react to the PAS/AB stains. It contained the characteristic lamellae (Figure 5F, Table 3) called “baffle plates” in the spinneret region and “plateau projections” extending toward the lumen; each lamella was surrounded by two short epithelial folds (spinnerets) that had two ciliated baffle plates. This region was composed of secretory and ciliated cells, and the presence of blood vessels, between tubules (Figure 5G, Table 4), was consistent, especially in the distal part near the smooth muscle tissue. The brown material was detected only between the secretory tubules of the baffle zone (Figure 5G). Finally, the terminal zone (Table 2) did not contain real lamellae (Figure 5H,I), and it consisted of serous and mucous tubules. The luminal tubular glands were a mixture of serous tubules (PAS-/AB-) with small granules and mucous tubules (AB+) with large vesicle cells (Figure 5H,I). Some gland tubules had a mixture of the two types of mucins, showing an apical region stained in slight blue (AB+), while the remaining were serous and similar to those of the baffle zone (Figure 5H,I).

##### Presence of Sperm

Sperm was detected in all examined species (Figure 6). In particular, unaggregated sperm was principally found within the lamellae (Figure 6A,D,G) and tubules (Figure 6B,E,H) of the baffle zone. Aggregated sperm bundles were found within an invagination between the terminal zone and the distal oviduct only in *C. monstrosa* (Figure 6C).

In *G. melastomus*, instead, bundles of sperm involved in acid mucopolysaccharides (AB+) were observed in the secretory tubules of the terminal zone (Figure 6F).

#### 3.2.2. Viviparous Species

##### *Centrophorus* *uyato*

In this species, it is possible to discern the club, papillary and baffle zones mainly on the basis of the luminal profile (Figure 2D). The club zone (Table 2) presented the typical club-shaped lamellae (Figure 7A, Table 3), and the glandular tubules, rounded in shape both near the lamellae and in the deep recesses of the gland (Table 4), strongly reacted to the AB/PAS staining, showing an intense purple coloration (Figure 7B). The papillary zone (AB+/PAS+), similar in extension to CZ (Table 2), possessed tubules (Table 4) showing different shapes: near the lamellae they were rounded (Table 4; Figure 7C), while towards the connective tissue they assumed a more elongated shape (Figure 7D). The baffle zone occupied the most part of the OG (Table 2) and showed very elongated lamellae (Figure 7E, Table 3) and tubules (Figure 7F, Table 4) reacting to the AB/PAS stain (AB+/PAS+). The terminal zone (Table 2) did not show very developed lamellae (Figure 7G, Table 3). It displayed several elongated tubules (Table 4) that showed affinity for PAS (PAS+, Figure 7H) and shorter ones (near the distal oviduct) with an affinity AB+/PAS+ (Figure 7G).

##### *Dalatias* *licha*

The OG of *D. licha* (Figure 2E) showed long lamellae in club, papillary and baffle zones (Figure 8A,C,E, Table 3); instead, the terminal zone possessed short lamellae (Figure 8G, Table 3). Club and papillary zones showed rounded and elongated tubules (Table 4), respectively, with a simple ciliated cuboidal epithelium (Figure 8B,D). More specifically, club zone tubular glands reacted slightly to PAS (PAS+) Figure 8D), while the papillary zone showed a strong affinity for neutral mucins (PAS+) (Figure 8B). In mature females, a great amount of secretory material was found in both the club and papillary zones (Figure 8B,D). The baffle and terminal zones (Table 2) showed secretory tubules with a simple ciliated columnar epithelium (Figure 8F,H). In particular, the baffle zone also showed secretory material within the tubules and the lamellae with PAS affinity (PAS+) (Figure 8E,F). The terminal zone showed few tubules (Table 4) in respect to the other zones (Figure 8H).

##### *Heptranchias* *perlo*

The only two OGs analyzed (belonging to regressing females) showed the typical zonation displayed in the other viviparous glands (Figure 2F) with club and papillary occupying 27.8% and 23.1% of the entire surface, respectively (Table 2). These zones did not show specifical affinity for dyes, but inside the secretory tubules (especially those near the connective tissue), secretory granules stained in magenta (PAS+) were detected (Figure 9B,D). Neither the baffle nor the terminal zone showed a reaction to the staining method (Figure 9E–H). The baffle zone (occupying 38.5% of the gland, Table 2) showed long plateau projection and baffle plates (Table 3; Figure 9E). Secretory granules stained in magenta (PAS+) were found within the tubules (Figure 9F) and the lamellae. The terminal zone was the smallest one (10.6% of the OG surface); it showed scantier and small tubules and few lamellae (Table 3 and Table 4; Figure 9G–H) in respect to the other zones.

##### *Oxynotus* *centrina*

The microscopic distinction of the zones of *O. centrina* mature OG appeared less marked in respect to oviparous species (Figure 2). In females capable of reproducing, tubular glands of the club and papillary zones showed affinity for PAS+, as the epithelium of their lamellae (Figure 10A–D). The baffle zone consisted of many elongated tubules rich in secretory material (PAS+ secretions), while the epithelium of lamellae was positive to AB (Figure 10E,F). The terminal zone consisted of elongate tubules PAS+ that ran adjacent to the baffle zone and terminated in recesses beyond the peripheral tubules of the baffle zone (Figure 10G,H). Accumulation of secretory material, PAS+, in the epithelium of secretory tubules and lamellae was observed, while brown material was detected between the secretory tubules (Figure 11E).

##### Presence of Sperm

All examined species showed the presence of sperm storage tubules (SSTs), with the exception of *C. uyato*, in which spermatozoa were observed only in secretory tubules (Figure 11A). SSTs were clearly distinguishable from secretory ones in dimension and structure being composed of a simple cuboidal epithelium with ciliated and secretory cells, containing sperm without any specifical pattern in their disposition, involved in a PAS+ matrix in its lumen (Figure 11). These structures were usually located in the terminal zone towards the lumen and/or near the distal oviduct (Figure 11B,D–F) in all species, except for *O. centrina* and *D. licha*, which also showed SSTs in the deep recesses of the terminal zone. Moreover, in *D. licha*, sperm were also observed inside the secretory tubules near the serosa of the OG (Figure 11C). Most females assigned to the capable of reproducing, pregnant, regressing and regenerating stages presented sperm within the SSTs (Figure 11A–E). Only in one *O. centrina* female in mid-pregnancy (stage 3C) were SSTs found empty of sperm (Figure 11F).

## 4. Discussion

To the best of our knowledge, in this work, for the first time, information on the OG morphology and microarchitecture of the holocephalan *C. monstrosa* and sharks belonging to the orders Hexanchiformes (*H. perlo*) and Squaliformes (*C. uyato*, *D. licha* and *O. centrina*) distributed in the Mediterranean [40,41] has been reported. The structure and organization of this organ are retained in all examined species, independently of their reproductive modality. However, the reproductive relevance of each glandular region, the secretion types and the conservation of sperm varied among them.

In the three oviparous species (*C. monstrosa*, *G. melastomus* and *R. polystigma*), the OGs were greatly extended since they produce not only the jellies that envelop the egg, but also an external capsule and all its ornamentations, reaching the maximum dimensions during the mature phase (e.g., [15,22,25,42]). The considerable enlargement of the OG in these species was due to the most extensive baffle zone (highly specialized in producing the egg case), which accounted for up to 62–66% of the total gland volume according to other oviparous species analyzed (e.g., [7,14,15,24]), differently from the three other zones which accounted for the remaining gland volume with the terminal zone occupying the smallest part. The morphometric analysis conducted in this study seemed to confirm this pattern. In fact, in *G. melastomus* and *R. polystigma*, lamellae and glandular tubules of the baffle zone were the most developed. On the contrary, in *C. monstrosa*, the most developed tubules were those of the terminal zone. This fact is probably due to the peculiar structure of its egg case characterized by a tear-drop shape with an elongated peduncle ending in a filament, produced by the terminal zone, that the species inserts into the substrate [42]. In addition, the microscopic structure of the *C. monstrosa* OG differed from other holocephalans such as the Australian ghostshark *Callorhinchus milii* [43]. The mucous secretions produced from the latter displayed an affinity for Alcian blue (acid mucopolysaccharides), while *C. monstrosa* produced neutral mucopolysaccharides (strong affinity for PAS). This different pattern could be explained by the different substrata in which the two species lay the egg cases. In fact, *C. milii* inhabits environments at depths <200 m [44], while *C. monstrosa* is distributed exclusively in bathyal ecosystems [40,41,45]. Regarding the other zones of the OG, club and papillary displayed a similar secretion pattern and occupied an analogous surface in the three species, except the “caudal-most papillary” (an intersection area between papillary and baffle zone) being found present in *C. monstrosa* and *R. polystigma* as in other oviparous species [14,15,23,45,46] and not detected in *G. melastomus*. The production of mucus that surrounds the fertilized egg and covers the interior walls of the different egg cases could be the cause of the structural differences [17,43].

The presence of spermatozoa was detected in all oviparous species. More specifically, in *R. polystigma* and *C. monstrosa*, disaggregated individual spermatozoa were found exclusively in the lamellae and tubules of the baffle zone. This fact could be the result of a recent mating episode or due to a “short-term storage” as reported for other oviparous species, mainly skates (e.g., [14,15,17]). In *R. polystigma*, as reported by Porcu et al. [47], mating occurs in coastal waters (<100 m) where populations are sexually segregated, behavior that does not necessarily require sperm storage. The observation of disaggregated sperm in *C. monstrosa*, instead, seemed to be unusual considering its deep bathymetric distribution. However, the presence of a great amount of sperm involved in an acid mucous matrix, detected in the distal oviduct, could be a clear sign of keeping spermatozoa, as also detected in other holocephalans [43,48,49]. Further investigations should be performed to verify this statement. The blackmouth catshark *G. melastomus* seemed to be the only oviparous species analyzed that retains sperm for a long time (long-term storage) showing sperm bundles involved in an AB+ matrix found in secretory glands of the terminal zone at different maturity stages (spawning capable, actively spawning and regressing). Long-term storage could be advantageous for deep species as *G. melastomus*, in which evidence of segregation by sex with females more frequent at greater depths [45] was found. In this way, the species would ensure fertilization occurs asynchronously with mating events [18,27]. In this sense, sperm storage is believed to be a particularly important strategy for oviparous species, such as catsharks, where the timing between mating and ovulation is often decoupled [50]. 

Differently from the oviparous species, viviparous species (*C. uyato*, *D. licha*, *H. perlo* and *O. centrina*) showed a smaller and less developed OG with respect to the body dimensions reached by these sharks. The volume breakdown of the baffle zone observed in the viviparous sharks (37–57% of the total gland) could be justified by the production of a thin, not durable, candle case in respect to a sturdy egg case produced by the oviparous species [1]. Some viviparous sharks such as *Centroscymnus coelolepis* [19] showed in the OG a distinct dark band, externally visible, that corresponds to the baffle zone and is probably associated with the production of the candle case. We found this brown coloration in adult OG females of *O. centrina*, and we can affirm surely that this species produces a candle case.

Club and papillary zone displayed a similar pattern already found in other viviparous species (e.g., [16,18,19,20]). Furthermore, our results highlight that the terminal zone was more developed in viviparous species (max 21.9% of total gland dimension) than in oviparous ones (max 9%). A possible explanation of this fact is the presence of sperm storage tubules (SSTs), specialized structures where spermatozoa are stored for a long time, which probably require a more extended surface. All the analyzed species presented SSTs in the terminal zone and in the posterior oviduct, suggesting long-term storage [28]. Only *C. uyato* did not seem to have these structures, even if bundles of sperm were found within terminal zone tubules in the deep recesses as also found in *D. licha* and other viviparous species [16,18]. The location of the SSTs, found in all mature stages, might give an indication of the elapsed time for mating. The finding of SSTs near the surface of the OG could indicate a recent formation, while those observed in the deep recesses (with more secretory vesicles) near the connective tissue could represent a longer retention of sperm as observed in *C. coelolepis* [19]. 

Long-term storage, in association with the presence of SSTs, here reported for the first time for the deep-sea sharks analyzed, represents a phenomenon clearly linked to the viviparity, both aplacental and placental (e.g., [18,19,20,51]). It is an advantageous mechanism that could bring several benefits: (i) ensuring the fertilization in systems where males and females are, or can be, largely solitary or separated [18,27,52] with females that may migrate away from males to more favorable habitats to release the litter [53,54,55]; (ii) guaranteeing the reproductive success and ensuring fertilization for copulation, fertilization and parturition [18,19,56], especially in species displaying low population densities such as the species analyzed in this work [42,43]; (iii) allowing the female to maximize the genetic quality of offspring whilst also ensuring a maximum number of offspring [18,27]; and (iv) avoiding energetically expensive and potentially damaging multiple mating events and possible injury caused by aggressive male mating behavior [16,21,57,58,59]. Sperm storage could represent an adaptive response to shark mating behavior that may also have benefits in the relatively low productivity environment of the deep sea.

## 5. Conclusions

Information reported here represents a step to expand the knowledge concerning the reproductive biology of chondrichthyan species showing different reproductive modalities. In fact, understanding the entire reproductive processes of Chondrichthyes and, in particular, the sperm storage mechanism which influences the reproductive potential of a species will constitute an input to understand the behavior, ecology and population dynamics of these species. It would be useful to conduct analysis on the spatial distribution, especially for species showing long-term storage, to test if females migrate away from males to find optimal environments to release their offspring and to identify possible spawning grounds. 

## Figures and Tables

**Figure 1 animals-11-02653-f001:**
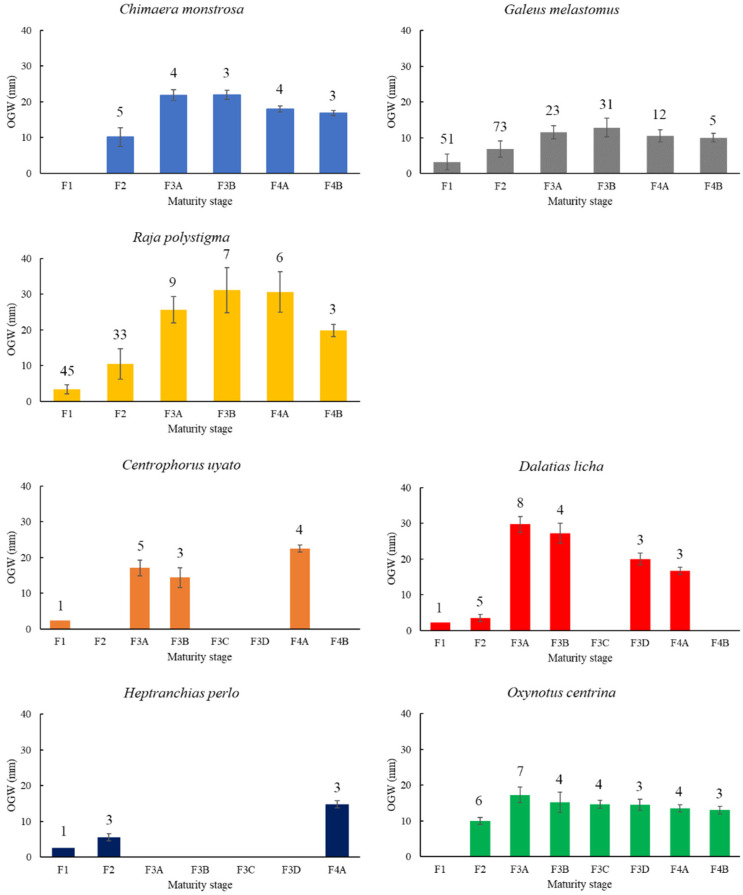
Changes of the oviducal gland width (OGW) through maturity stages in oviparous (*C. monstrosa*, *G. melastomus*, *R. polystigma*) and viviparous (*C. uyato*, *D. licha*, *H. perlo*, *O. centrina*) species. Above the bars, the number of the OGs analyzed is reported.

**Figure 2 animals-11-02653-f002:**
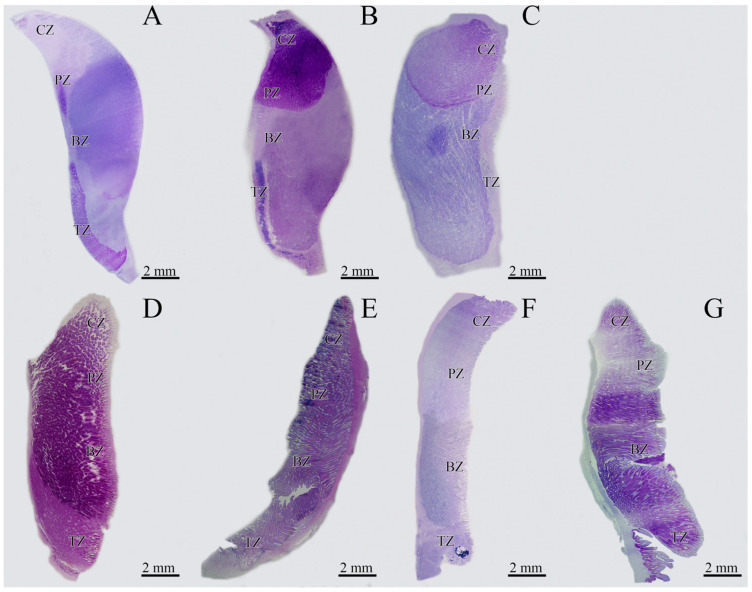
Sagittal section of the oviducal gland (OG) belonging to oviparous and viviparous species. (**A**) *C. monstrosa*; (**B**) *G. melastomus*; (**C**) *R. polystigma*; (**D**) *C. uyato*; (**E**) *D. licha*; (**F**) *H. perlo*; (**G**) *O. centrina*. BZ, baffle zone; CZ, club zone; PZ, papillary zone; TZ, terminal zone.

**Figure 3 animals-11-02653-f003:**
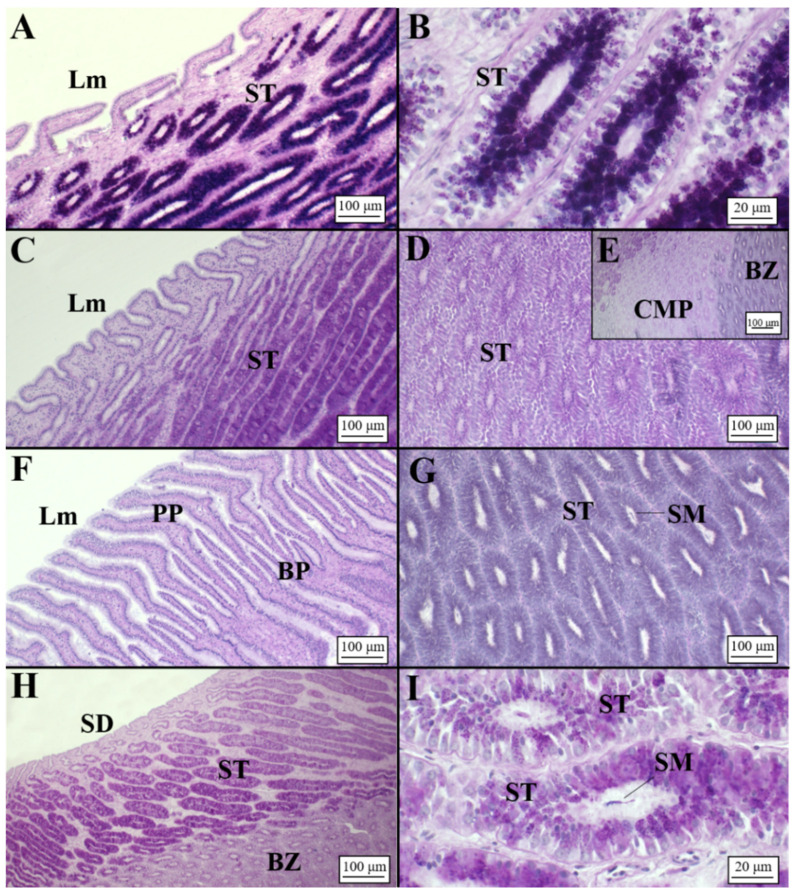
Microarchitecture of a *C. monstrosa* mature female (AB/PAS staining method applied to all slides). (**A**) Club-shaped lamellae and secretory tubules of the club zone. (**B**) High magnification of the club tubules having secretory granules stained AB+/PAS+ (deep purple color). (**C**) Digitiform lamellae of the papillary zone and secretory tubules near them showing a PAS+ affinity. (**D**) Secretory tubules of the papillary zone in the deep recesses: tubules are rounded showing a light affinity for PAS. (**E**) Transitional zone between papillary and baffle zone known as caudal-most papillary in which it is possible to observe a slight affinity only for PAS. (**F**) Baffle lamellae constituted by the typical plateau projections and baffle plates. (**G**) High magnification of the baffle zone tubules with secretory material distinguishable inside them and no affinity for dyes (AB-/PAS-). (**H**) Terminal zone showing secretory ducts opened towards the lumen. Tubules are elongated, showing an affinity for PAS (PAS+). (**I**) High magnification of the terminal zone secretory tubules in which the secretory material is visible. BP, baffle plates; BZ, baffle zone; CMP, caudal-most papillary; Lm, lamellae; PP, plateau projections; SM, secretory material; SD, secretory ducts; ST, secretory tubules.

**Figure 4 animals-11-02653-f004:**
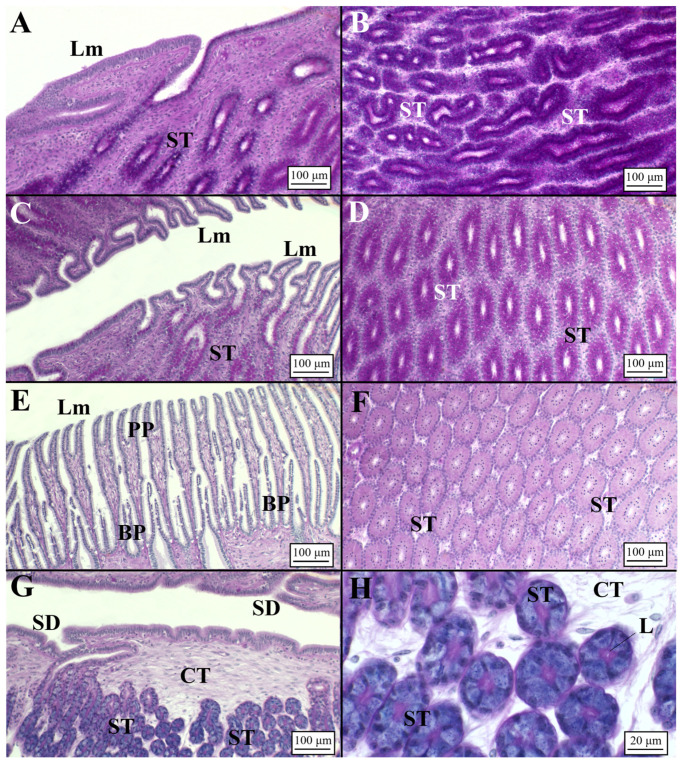
Microarchitecture of a *G. melastomus* mature female (AB/PAS staining method applied to all slides). (**A**) Club-shaped lamellae and secretory tubules of the club zone. (**B**) High magnification of the club tubules having secretory granules stained in AB+/PAS+ (purple color). (**C**) Digitiform lamellae of the papillary zone and secretory tubules near them showing a PAS+ affinity. (**D**) Secretory tubules of the papillary zone showing affinity for PAS. (**E**) Baffle lamellae constituted by the typical plateau projections and baffle plates. (**F**) High magnification of the baffle zone tubules showing no affinity for dyes (AB-/PAS-). (**G**) Terminal zone showing secretory ducts opened towards the lumen. Tubules are elongated, showing an affinity for acid mucins (AB+). (**H**) High magnification of the terminal zone secretory tubules showing an intense affinity for Alcian blue (AB+). BP, baffle plates; CT, connective tissue; L, lumen; Lm, lamellae; PP, plateau projections; SM, secretory material; SD, secretory ducts; ST, secretory tubules.

**Figure 5 animals-11-02653-f005:**
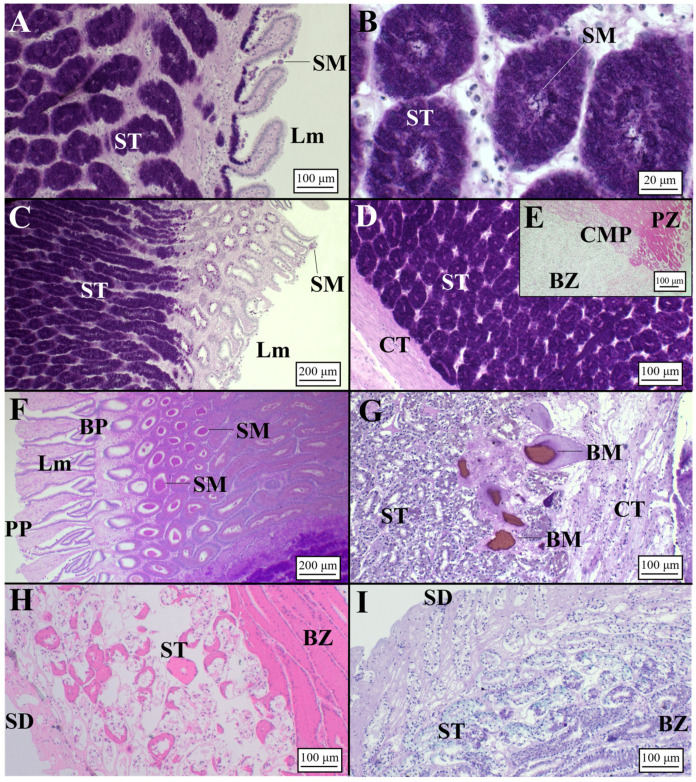
Microarchitecture of a *R. polystigma* mature female (AB/PAS staining method applied to all slides, except to (**E**,**H**) stained in (**H**,**E**)). (**A**) Club-shaped lamellae with secretory material found within them and secretory tubules of the club zone. (**B**) High magnification of the club tubules stained AB+/PAS+ (deep purple color) having secretory material inside them. (**C**) Digitiform lamellae of the papillary zone with secretory material within them and secretory tubules. (**D**) Secretory tubules of the papillary zone showing affinity for AB+/PAS+ (deep purple color). (**E**) Transitional zone between papillary and baffle zone (caudal-most papillary) in which it is possible to observe a slight affinity for dyes. (**F**) Baffle lamellae constituted by the typical plateau projections and baffle plates and tubules of the baffle zone containing a great amount of secretory material. (**G**) Deep recesses of the baffle zone near the connective tissue in which tubules do not show affinity for dyes (AB-/PAS-). Brown material was detected. (**H**) Terminal zone showing secretory ducts opened towards the lumen. Tubules are rounded, showing a mixture of mucins. (**I**) Terminal zone secretory tubules showing a slight affinity for Alcian blue (AB+) and also for PAS (PAS+). BM, brown material; BP, baffle plates; BZ, baffle zone; CMP, caudal-most papillary; CT, connective tissue; Lm, lamellae; PP, plateau projections; SM, secretory material; SD, secretory ducts; ST, secretory tubules.

**Figure 6 animals-11-02653-f006:**
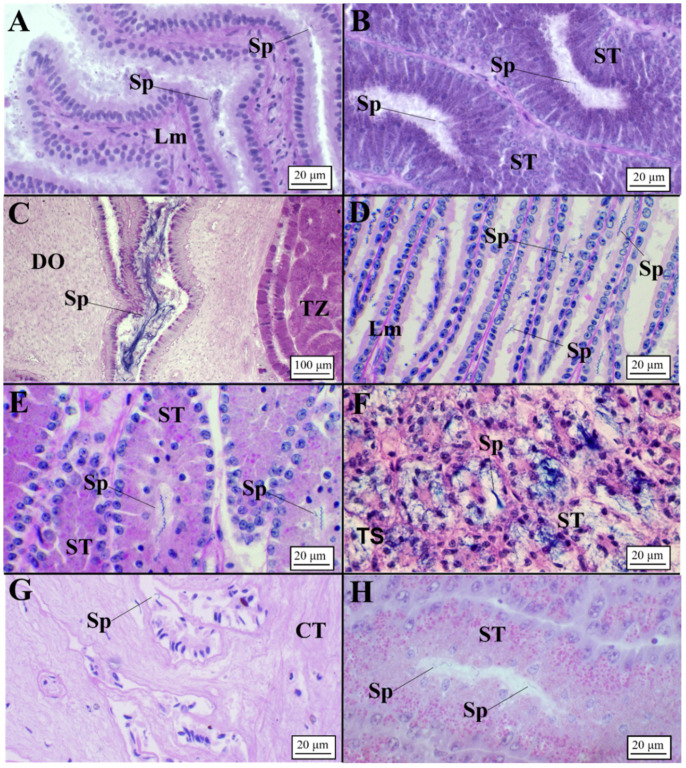
Presence of sperm in oviparous species (AB/PAS staining method applied to all slides). (**A**,**B**) Unaggregated sperm found between lamellae (**A**) and tubules (**B**) of the baffle zone in *C. monstrosa*. (**C**) Aggregated bundles of sperm found between the terminal zone and the distal oviduct in *C. monstrosa*. (**D**,**E**) Unaggregated sperm found between lamellae (**D**) and tubules (**E**) of the baffle zone in *G. melastomus*. (**F**) Bundles of sperm involved in AB+ matrix found in terminal zone tubules in *G. melastomus*. (**G**,**H**) Unaggregated sperm found between lamellae (**G**) and tubules (**H**) of the baffle zone in *R. polystigma*. CT, connective tissue; DO, distal oviduct; Lm, lamellae; Sp, sperm; ST, secretory tubules; TZ, terminal zone.

**Figure 7 animals-11-02653-f007:**
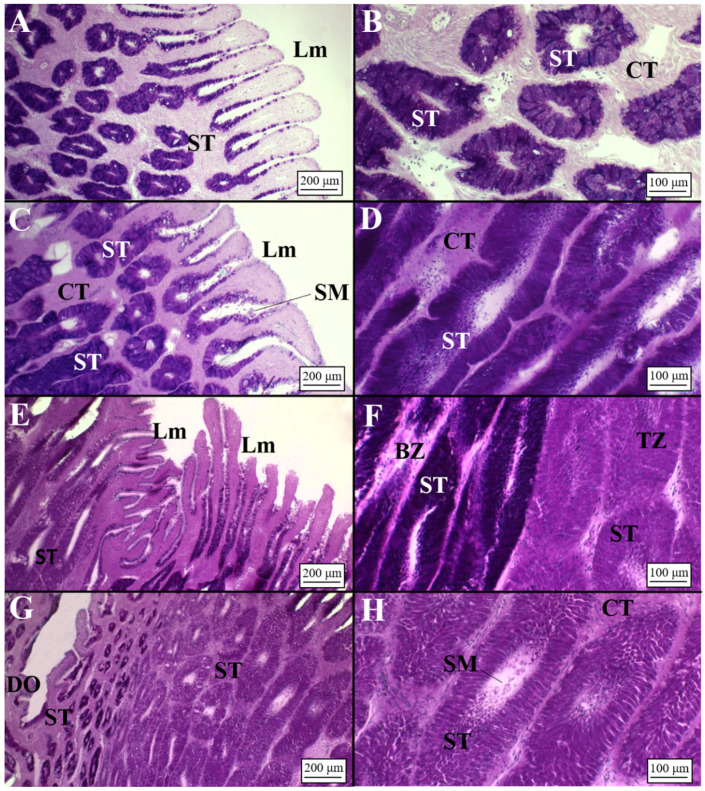
Microarchitecture of a *C. uyato* mature female (AB/PAS staining method applied to all slides). (**A**) Club-shaped lamellae and secretory tubules of the club zone showing AB+/PAS+ affinity. It is possible to observe the affinity for dyes also in the epithelium bordering the lamellae. (**B**) High magnification of the club tubules having different shapes (rounded and elliptical) stained AB+/PAS+ (purple color). (**C**) Digitiform lamellae of the papillary zone and secretory tubules near them showing AB+/PAS+ affinity. The epithelium bordering the lamellae reacted to AB/PAS staining and secretory material detected within them. Tubules near the lamellae have a rounded shape. (**D**) Elongated secretory tubules of the papillary zone showing affinity for AB+/PAS+. (**E**) Baffle lamellae that are thin and long showing affinity for dyes in the bordering epithelium. (**F**) High magnification of the baffle zone tubules bordering the terminal zone. Tubules of the baffle zone strongly reacted to AB/PAS. (**G**) Terminal zone showing short lamellae. The majority of tubules are elongated, showing an affinity for neutral mucins (PAS+), and only a few of them, near the distal oviduct, are smaller and displayed an affinity for AB+/PAS+. (**H**) High magnification of the terminal zone secretory tubules showing an intense affinity for PAS (PAS+) and secretory material within them. BZ, baffle zone; CT, connective tissue; DO, distal oviduct; Lm, lamellae; SM, secretory material; SD, secretory ducts; ST, secretory tubules; TZ, terminal zone.

**Figure 8 animals-11-02653-f008:**
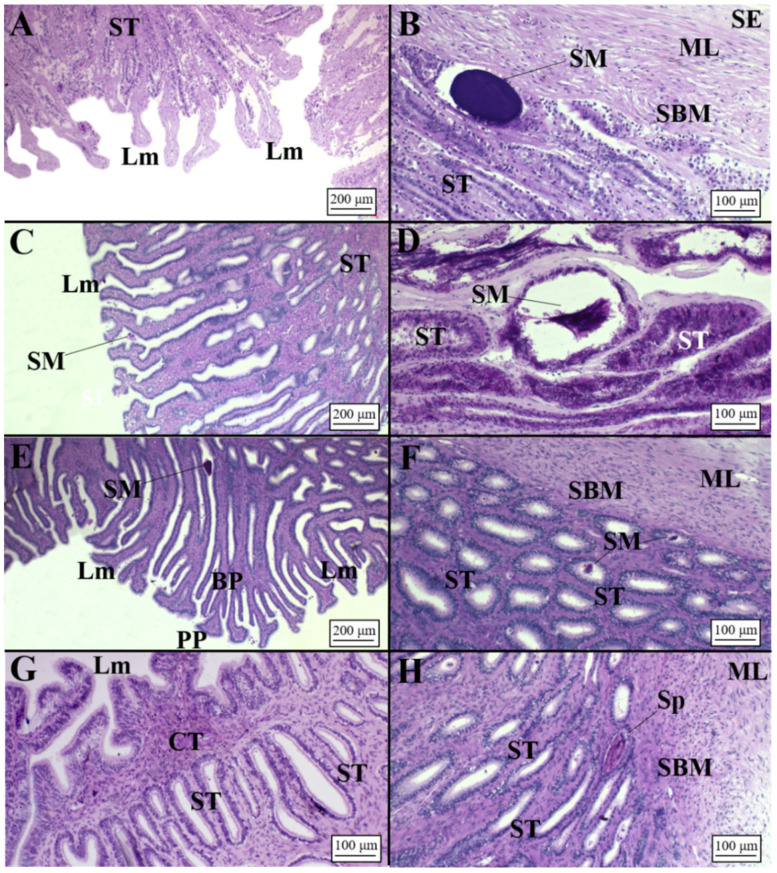
Microarchitecture of a *D. licha* mature female (AB/PAS staining method applied to all slides). (**A**) Long lamellae of the club zone showing a slight affinity for PAS+. (**B**) Club zone secretory tubules, rich in secretory material, located in the deep recesses. (**C**) Papillary lamellae and secretory tubules near them. Secretory material was detected between the lamellae. The epithelium bordering the lamellae reacted to AB/PAS staining, and secretory material was detected within them. Tubules near the lamellae have a rounded shape. (**D**) Secretory tubules in the deep recesses of the OG showing affinity for PAS. A great amount of secretory material was found inside them. (**E**) Thin and long baffle lamellae composed of baffle plates and plateau projections. (**F**) High magnification of the baffle zone tubules bordering the connective tissue. Secretory material was detected. (**G**) Terminal zone showing short lamellae and elongated secretory tubules composed of secretory ciliated cells near them. (**H**) Deep recesses of the terminal zone in which sperm, involved in a PAS+ matrix, were detected. BP, baffle plates; CT, connective tissue; Lm, lamellae; ML, muscular layer; PP, plateau projections; SBM, submucosa; SD, secretory ducts; SE, serosa; SM, secretory material; Sp, sperm; ST, secretory tubules.

**Figure 9 animals-11-02653-f009:**
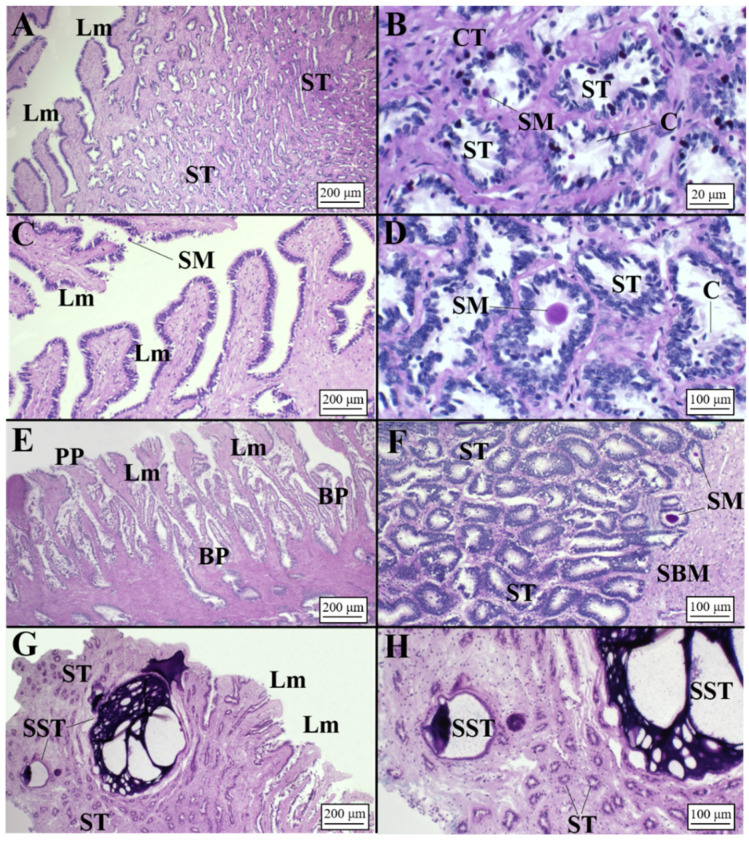
Microarchitecture of an *H. perlo* regressing female (AB/PAS staining method applied to all slides). (**A**) Club zone lamellae and rounded secretory tubules without any affinity for dyes. (**B**) High magnification of club zone glandular tubules with secretory material weakly PAS+. (**C**) Lamellae of the papillary zone. (**D**) Detail of glandular tubules with secretory material PAS+. (**E**) Plateau projections and baffle plates of the baffle zone. (**F**) Tubular glands of the baffle zone. The secretory tubules found in the deep recesses showed affinity for PAS. (**G**) Lamellae and tubular glands of the terminal zone. (**H**) Terminal zone in which sperm storage tubules are visible. BP, baffle plates; C, cilia; CT, connective tissue; LM, lamellae; PP, plateau projections; SBM, submucosa; SM, secretory material; SST, sperm storage tubules; ST, secretory tubules.

**Figure 10 animals-11-02653-f010:**
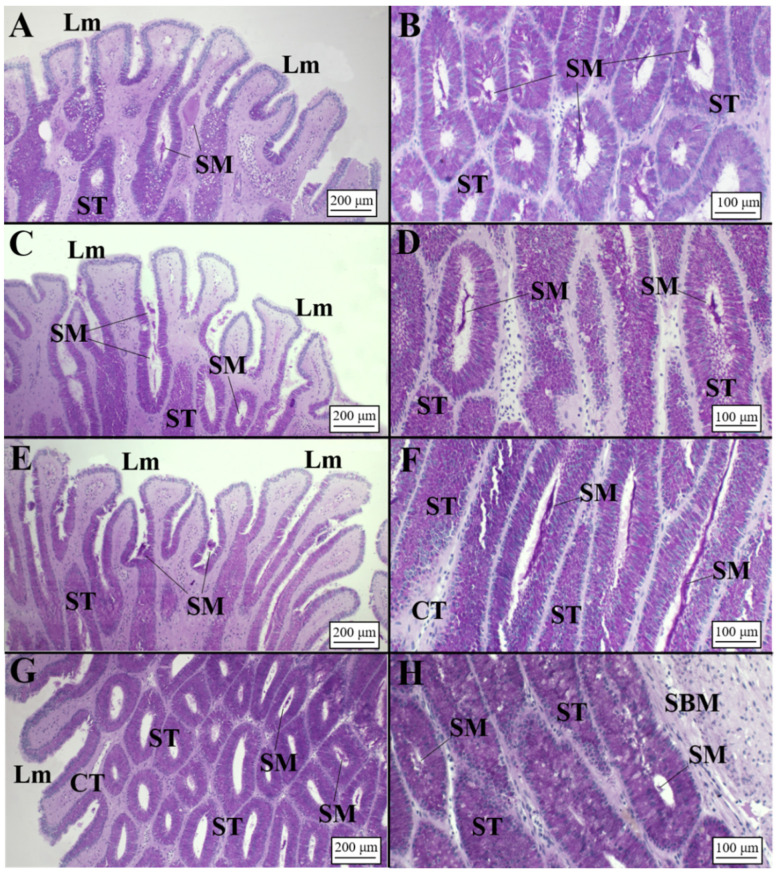
Microarchitecture of an *O. centrina* mature female (AB/PAS staining method applied to all slides). (**A**) Lamellae of the club zone. Secretory material (PAS+) is visible inside and between lamellae. (**B**) Secretory tubules of the club zone full of secretory material PAS+. (**C**) Lamellae of the papillary zone. Secretory material (PAS+) is visible between lamellae and inside tubules. (**D**) Secretory tubules of the baffle zone full of secretory material (PAS+). (**E**) Lamellae of the baffle zone with secretory material (PAS+) between them. (**F**) High magnification of elongated secretory tubules in the deep recesses of the baffle zone. (**G**) Terminal zone with lamellae visible. Secretory tubules showed PAS+ secretory material inside. (**H**) High magnification of secretory tubules of the terminal zone. CT, connective tissue; Lm, lamellae; SBM, submucosa; SM, secretory material; ST, secretory tubules.

**Figure 11 animals-11-02653-f011:**
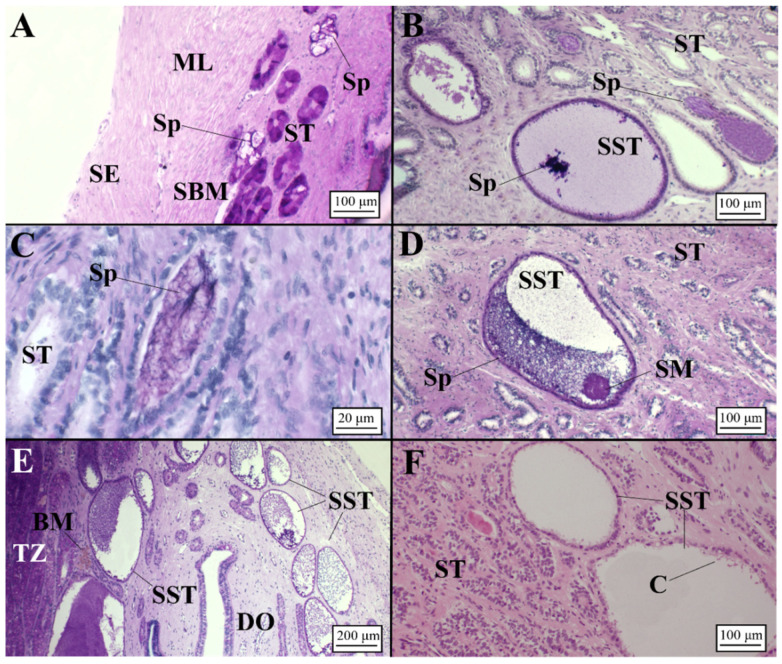
Presence of sperm in viviparous species (AB/PAS staining method applied to all slides, except to F stained in H&E). (**A**) Sperm found inside the secretory tubules in the terminal zone near the serosa, which enclosed the gland in *C. uyato*. (**B**) Detail of a sperm storage tubule containing spermatozoa, involved in PAS+ matrix, located in the deep recesses of the terminal zone and (**C**) high magnification of a secretory tubule with sperm in the terminal zone, near the distal oviduct, in *D. licha*. (**D**) High magnification of a sperm storage tubule with sperm and secretory material visible located in the terminal zone of *H. perlo*. (**E**) Several sperm storage tubules full of sperm located in the terminal zone and near the distal oviduct and (**F**) empty sperm storage tubules, bordered by a simple cuboidal epithelium with cilia inside, found in mid-pregnancy female of *O. centrina*. C, cilia; DO, distal oviduct; ML, muscular layer; SBM, submucosa; SE, serosa; SM, secretory material; Sp, sperm; SST, sperm storage tubule; ST, secretory tubule; TZ, terminal zone.

**Table 1 animals-11-02653-t001:** List of the selected chondrichthyan species. Reproductive mode, depth, size ranges and IUCN status in the Mediterranean are presented.

Reproduction Mode	Order	Family	Species	Depth Range (m)	Size Range (cm)	IUCN Status
Oviparity	Carcharhiniformes	Pentanchidae	*Galeus melastomus* (Rafinesque, 1810)	100–730	6.9–89.7 ^†^	LC
	Chimaeriformes	Chimaeridae	*Chimaera monstrosa* (Linnaeus, 1758)	320–1132	3.6–27.8 *	NT
	Rajiformes	Rajidae	*Raja polystigma* (Regan, 1923)	33–600	15.6–59.3 ^†^	LC
Viviparity (yolk-sac)	Hexanchiformes	Hexanchidae	*Heptranchias perlo* (Bonnaterre, 1788)	336–600	54.5–113.3 ^†^	DD
	Squaliformes	Centrophoridae	*Centrophorus uyato* (Rafinesque, 1810)	445–600	38.7–106.5 ^†^	CR
		Dalatidae	*Dalatias licha* (Bonnaterre, 1788)	300–800	26.2–108.6 ^†^	VU
		Oxynotidae	*Oxynotus centrina* (Linnaeus, 1758)	175–600	25.6–78.2 ^†^	CR

CR, critically endangered; DD, data deficient; LC, least concern; NT, near threatened; VU, vulnerable. * AL, anal length. ^†^ TL, total length.

**Table 2 animals-11-02653-t002:** Oviducal gland area expressed in mm^2^ and % of the total gland for the four zones in the oviparous (*C. monstrosa*, *G. melastomus*, *R. polystigma*) and viviparous (*C. uyato*, *D. licha*, *H. perlo*, *O. centrina*) species examined.

Species	Club		Papillary		Baffle		Terminal	
	mm^2^	%	mm^2^	%	mm^2^	%	mm^2^	%
*C. monstrosa*	12.8–13.9	13.2–13.6	13.0–14.2	13.3–13.8	63.0–67.2	63.1–66.4	8.5–9.3	8.2–9.0
*G. melastomus*	10.1–11.2	18.0–19.5	4.9–6.3	9.9–10.8	34.7–35.9	60.1–65.0	4.1–4.9	7.8–8.5
*R. polystigma*	7.2–8.4	15.0–16.2	7.3–8.2	15.0–16.2	28.6–34.3	61.1–66.7	1.9–2.3	4.2–4.8
*C. uyato*	18.2–21.9	20.6–220	23.5	23.2	38.2	37.7	18.1	17.9
*D. licha*	3.8–5.1	10.6–11.9	9.6–10.7	24.5–26.0	21.5–22.9	55.1–56.9	1.9–2.9	5.8–6.9
*H. perlo*	14.9	27.8	12.4	23.1	20.7	38.5	5.6	10.6
*O. centrina*	9.9–11.0	13.2–15.1	8.6–10.4	12.2–14.0	35.4–36.9	48.5–50.8	14.4–15.7	20.4–21.9

**Table 3 animals-11-02653-t003:** Number, minimum, maximum, mean length (µm) and standard deviation (S.D.) of lamellae belonging to different OG zones in oviparous (*C. monstrosa*, *G. melastomus*, *R. polystigma*) and viviparous (*C. uyato*, *D. licha*, *H. perlo*, *O. centrina*) species.

Species	Club			Papillary			Baffle		
	N	Mean ± S.D.	Min–Max	N	Mean ± S.D.	Min–Max	N	Mean ± S.D.	Min–Max
*C. monstrosa*	37	94.3 ± 45.5	56.3–235.4	33	119.6 ± 19.9	89.5–169.5	34	407.7 ± 88.0	264.5–537.8
*G. melastomus*	15	177.7 ± 34.4	128.9–212.7	8	108.8 ± 22.3	63.3–141.2	28	506.6 ± 29.5	476.3–550.9
*R. polistygma*	14	296.9 ± 165.3	101.4–786.9	12	252.8 ± 59.7	158.5–373.1	22	584.3 ± 105.5	399.4–792.4
*C. uyato*	31	857.9 ± 69.0	725.7–982.2	14	828.8 ± 207.6	503.7–1407.1	43	304.6 ± 151.7	136.8–572.4
*D. licha*	17	1555.5 ± 233.2	1046.6–1701.0	26	1804.4 ± 38.2	1756.2–1892.3	56	1013.0 ± 452.4	191.4–1962.8
*H. perlo*	19	547.7 ± 174.5	193.2–803.0	21	442.6 ± 88.9	307.8–575.6	22	982.8 ± 327.9	534.7–1459.2
*O. centrina*	16	532.4 ± 92.5	368.5–617.4	16	695.8 ± 180.4	519.0–1167.4	44	888.0 ± 200.7	600.0–1239.6

**Table 4 animals-11-02653-t004:** Minimum, maximum, mean length (µm) and standard deviation (S.D.) of secretory tubules belonging to different OG zones in oviparous (*C. monstrosa*, *G. melastomus*, *R. polystigma*) and viviparous (*C. uyato*, *D. licha*, *H. perlo*, *O. centrina*) species.

Species	Club		Papillary		Baffle		Terminal	
	Min–Max	Mean ± S.D.	Min–Max	Mean ± S.D.	Min–Max	Mean ± S.D.	Min–Max	Mean ± S.D.
*C. monstrosa*	71.0–354.5	193.6 ± 67.8	82.3–566.6	228.5 ± 86.3	103.8–443.6	191.5 ± 60.3	89.0–614.2	267.9 ± 115.5
*G. melastomus*	65.1–332.4	132.0 ± 82.7	126.1–353.9	175.4 ± 46.4	174.0–429.1	262.4 ± 89.0	39.3–140.8	73.9 ± 22.1
*R. polistygma*	64.9–408.2	179.1 ± 75.1	64.4–483.0	169.9 ± 85.1	101.8–768.4	223.8 ± 106.9	52.0–428.6	119.0 ± 84.4
*C. uyato*	136.9–509.8	279.8 ± 70.8	140.7–1022.5	353.0 ± 122.9	190.8–1027.3	513.5 ± 179.8	84.2–1141.9	174.5 ± 166.2
*D. licha*	58.1–568.8	194.7 ± 108.7	89.4–683.8	269.2 ± 137.3	59.7–539.7	198.7 ± 84.7	63.0–434.1	176.7 ± 88.8
*H. perlo*	43.1–496.5	161.7	58.0–335.5	117.5 ± 41.7	60.2–343.1	128.9 ± 60.4	30.9–198.1	71.59 ± 36.0
*O. centrina*	173.3–1133.6	432.9 ± 195.9	222.3–971.0	444.0 ± 177.9	143.6–1230.9	494.6 ± 264.4	281.9–869.6	461.8 ± 130.0

## Data Availability

The data presented in this study are available on request from the corresponding authors.

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
