# Peer review of "A Taxonomic Survey of Female Oviducal Glands in Chondrichthyes: A Comparative Overview of Microanatomy in the Two Reproductive Modes"

_animals, 2021, doi:10.3390/ani11092653_

Round 1

Reviewer 1 Report

My only observation is that the scope of the results should be limited or explained how the results of this study can be incorporated for the evaluation of the population dynamics or conservation and management actions.

In Table 1 it says, "List of chondrichthyan species caught during the sampling". The question is, are only those species caught, or were only those species selected for this study?

Author Response

Reviewer 1

We would like to thank the reviewer 1 for his/her thoughtful comments and we have no doubt that the reviewer’s suggestions will improve our manuscript.

Reviewer 2 Report

General comments:

The paper is interesting and provide novel information on the development of the oviducal gland in Chondrichthyes. My biggest concern with the manuscript is the push to try to show that this information is very important for the conservation and management of the species, too much relevance into the species IUCN categorisation. I do not think it is the right scope. I think the work of showing the difference OG in different reproductive modes is enough.

If the authors want to push the scope of this manuscript into conservation and management plans, then much more explanation need to be provided. For example, will the results of this manuscript really and truly be used in the management plans for any of these species in Europe? How? How this information is important for stock assessment? Size at maturity, the reproductive cycle and where the species reproduce is important in term of reproduction for the management plans, the rest could be added but it is not important really.

Would be different if the authors focus on active conservation, re population of the species, then it is important to know if the species storage sperm for captive for breeding programs, but for management policies, not really.

Introduction

The introduction has all the necessary information to understand the manuscript, but again I strongly recommend the authors to review the true link between the manuscript results and the conservation outputs and outcomes. See below for detailed comments:

Line 43: What the authors are referring to “small number of highly developed offspring” Small comparative to what? Also, for example around 300 embryos were found in whale shark uterus, not a very small number.

Line 49-54: I would recommend the authors to review the classification of reproductive modes on chondrichthyans and look for newer references. The proper terminology to further separate viviparous species related to nutrition, is lecithotrophic or matrotrophic species. A lecithotrophic terminology is more accurate concept to use than aplacental.

Line 77-78: add a reference on the concept of sperm storage likely occur in most internally fertilizing animals. Also on line 78, do no start a new sentence with an abbreviation, is not proper English. Look throughout the manuscript for other sentences starting with abbreviation and change it accordingly.

Line 97-98: I think this sentence/concept should be revisited. The paper the authors are citing is 20 years old, much more work has been done since then.

Line 106-107: As previously mentioned, if the authors want to keep this sentence, I strongly recommend extending de concept and explain how the results of this manuscript will be incorporated into stock assessment models (are already the authors in conversation with the European policy makers, are the author able to give this information to the scientist producing the stock assessment models?), and how incorporating this information into the stock assessment models will contribute for the conservation of threatened species?.

Methodology:

By looking into the results section, particularly Figure 1, more information is needed on the methodology section.

One line 121-128: the author explained the reproductive stages, but on figure 1 there is an F before each stage, what the F is referring to?

On Line 128 the authors wrote that mature females from stage 3a to 4b were used for this study, but on figure 1 there are stages F1 and F2, which stages are those? More comments on figure 1 on the result section.

Results:

The result section is very well written, the figures and the text are very clear. I only have comments on figure 1.

Please add the sample sizes in figure 1 for each stage for each species. It is not clear on the figure 1 if there are significant differences between all values or some values. ANOVA will show significant differences, but further statistical tests need to be done to see where those significant differences are. For example, in figure 1 C. monstrosa, is there significant differences between all stages or between some stages?

Legend for figure 1: Considering replacing the change “evolution” for maybe “changes”. The oviducal gland is not evolving from one stage to another, is changing.

Discussion:

Line 458-462: I don’t think it is necessary to focus this first sentence on this study being the first one to describe the OG in threatened species in Europe. As I mention at the beginning of this review, the focus of this work should be on the OG description in different reproductive modes, not on the conservation.

Line 476-479: This sentence is too long. I would suggest the authors to re-write it to make it shorter or split in two.

Line 481-482: replace “this latter” for “the former” or “the latter”.

Line 484: replace “fix” by “lay”

Line 551-554: I would review this sentence, as said many times in other sections, explained more how this work will truly help conservation. Instead focusing on the conservation outcomes, I believe the authors have enough material to focus and develop more extensively the ideas expressed on lines 537- 550.  

Conclusions:

I believe these conclusions do not really summarise the main points of the manuscript. Unless the journal is asking for a conclusion section, I recommend the author to delete. If the authors must write a conclusion section, then I strongly recommend to the authors to re-think what are the main points learnt from this manuscript. It is not the hope that will help conservation but the profound description of the OG in species under different reproductive mode.

Author Response

Reviewer 2

We would like to thank the reviewer 1 for his/her thoughtful comments and we have no doubt that the reviewer’s suggestions will improve our manuscript.

Reviewer 3 Report

In this paper, the authors present a histological analysis of the oviducts of cartilaginous fishes. The morphology of the oviducts of cartilaginous fishes is still unclear due to the difficulty in obtaining samples. Therefore, the morphological information of the oviducts reported in this paper is considered to be very important.

This paper was well written and enjoyable to read. I do not see any need for correction, but I have minor comments as follows.

-There were many abbreviations in the paper, which made it difficult to read in some parts. A list of abbreviations should be provided in the MS.

-If possible, I would have liked to see sperm in the oviducts photographed under high magnification.

-I think that supplementary figures can be ordinary figures, but how about it?

Author Response

Reviewer 3

We would like to thank the reviewer 1 for his/her thoughtful comments and we have no doubt that the reviewer’s suggestions will improve our manuscript.
